# Discrete Diffusion for Reflective Vision-Language-Action Models in Autonomous Driving

**Pengxiang Li[1*‡], Yinan Zheng[2*‡], Yue Wang[1*], Huimin Wang[1†],**

**Hang Zhao[2], Jingjing Liu[2], Xianyuan Zhan[2], Kun Zhan[1], Xianpeng Lang[1]**

[1]LiAuto    [2]Tsinghua University

## Abstract

End-to-End (E2E) solutions have emerged as a mainstream approach for autonomous driving systems, with Vision-Language-Action (VLA) models representing a new paradigm that leverages pre-trained multimodal knowledge from Vision-Language Models (VLMs) to interpret and interact with complex real-world environments. However, these methods remain constrained by the limitations of imitation learning, which struggles to inherently encode physical rules during training. Existing approaches often rely on complex rule-based post-refinement, employ reinforcement learning that remains largely limited to simulation, or utilize diffusion guidance that requires computationally expensive gradient calculations. To address these challenges, we introduce *ReflectDrive*, a novel learning-based framework that integrates a reflection mechanism for safe trajectory generation via discrete diffusion. We first discretize the two-dimensional driving space to construct an action codebook, enabling the use of pre-trained Diffusion Language Models for planning tasks through fine-tuning. Central to our approach is a safety-aware reflection mechanism that performs iterative self-correction without gradient computation. Our method begins with goal-conditioned trajectory generation to model multi-modal driving behaviors. Based on this, we apply local search methods to identify unsafe tokens and determine feasible solutions, which then serve as safe anchors for inpainting-based regeneration. Evaluated on the NAVSIM benchmark, *ReflectDrive* demonstrates significant advantages in safety-critical trajectory generation, offering a scalable and reliable solution for autonomous driving systems.

## 1 Introduction

Autonomous driving (AD) is guiding the transportation industry toward a safer and more efficient future (Tampuu et al., 2020). Within this trend, End-to-End (E2E) systems (Hu et al., 2023; Chen et al., 2023) have emerged as the mainstream alternative to traditional modular designs (Bansal et al., 2018), which are prone to error accumulation between interdependent modules. They have also largely replaced rule-based methods (Fan et al., 2018; Treiber et al., 2000) that demand extensive human engineering effort. Meanwhile, Vision-Language-Action (VLA) models (Kim et al., 2024; Hwang et al., 2024) offer a new solution by incorporating pre-trained knowledge from Vision-Language Models (VLMs) (Hurst et al., 2024; Bai et al., 2025). Equipped with enhanced generalization capabilities, VLA models can interpret visual scenes and understand human instructions to directly output planning trajectories, thereby improving adaptability in challenging situations.

However, eixsting learning-based methods does not resolve the core challenge in imitation learning-based driving systems. Specifically, behavior cloning fails to inherently encode inviolable physical rules, such as collision avoidance or adherence to drivable areas (Lu et al., 2023). As a result, a generated trajectory may be highly probable under the model's distribution yet still violate critical safety constraints. Consequently, existing deployed solutions often rely on significant human priors, such as trajectory anchors (Li et al., 2024) or rule-based generated paths (Dauner et al., 2023).

These priors offer a reliable initial solution for the learning system, but they also necessitate substantial post-processing, particularly in complex scenarios. Concurrently, more advanced solutions are emerging. Some methods integrate reinforcement learning (Kaelbling et al., 1996; Kendall et al., 2019; Jaeger et al., 2025; Cusumano-Towner et al., 2025) with human-designed reward functions to enhance causal reasoning. However, most existing studies remain confined to the simulation level. From a deployment perspective, these approaches typically require unsafe online rollouts and suffer from training instability, especially in large-scale models (Zheng et al., 2024). Although guidance mechanisms in diffusion models provide a promising alternative by enabling controllable generation during inference (Zheng et al., 2025; Jiang et al., 2023; Zhong et al., 2023), they often experience slow sampling speeds due to gradient computations and are highly sensitive to parameter tuning, which can lead to numerical instability.

To address these challenges, we pioneer the use of discrete diffusion (Austin et al., 2021) for planning to meet the demand for verifiable and controllable E2E driving systems. A key advantage of this approach is its operation in a discrete action space, which facilitates the seamless incorporation of critical safety constraints through search, masking, and sampling techniques during trajectory generation. This results in a hybrid framework in which learned behaviors can be rigorously guided by prior knowledge, shifting away from black-box planning toward trustworthy and interpretable decision-making. Inspired by these insights, we propose *ReflectDrive*, a novel learning-based framework that integrates a reflection mechanism for safe trajectory generation via discrete diffusion. Specifically, we first discretize the two-dimensional driving space to construct a action codebook, enabling the representation of vehicle trajectories through discrete codebook embeddings. This representation allows us to leverage a pre-trained Diffusion Language Models (DLMs) (You et al., 2025; Nie et al., 2025) for planning tasks via fine-tuning. The approach facilitates parallel decoding and bidirectional feature fusion within a unified architecture that supports scalable training. Based on this fine-tuned model, our reflection mechanism begins with goal-conditioned generation, where the goal point guides the generation process to capture diverse multi-modal driving behaviors. Furthermore, the framework integrates safety metrics to evaluate the generated multi-modal trajectories. For unsafe waypoints, we perform a local search to identify a feasible solution, which then serves as a safe anchor token for trajectory inpainting. The entire process operates without gradient computation, enabling parallel generation and the injection of safety constraints during trajectory regeneration. Evaluations on the real-world autonomous driving benchmark NAVSIM (Dauner et al., 2024) demonstrate the feasibility of employing discrete diffusion for trajectory generation. Equipped with our reflection mechanism, *ReflectDrive* achieves near human-level closed-loop performance. Our contributions are summarized as follows:

- We pioneer the application of discrete diffusion for E2E autonomous driving trajectory generation and integrate it into a VLA model for scalable training.

- We introduce reflection mechanism, a novel inference-time guidance framework specifically designed for the denoising process in discrete diffusion, integrating external safety validation with efficient discrete token optimization.

- We evaluate our method on real-world driving benchmarks, proving that the framework can enforce hard safety constraints without compromising behavioral coherence.

## 2    RELATED WORK

**End-to-End Autonomous Driving.**    E2E methods (Hu et al., 2023; Chen et al., 2023) have emerged as a promising solution to largely replace rule-based approaches due to their superior scalability. Recently, VLA models (Hwang et al., 2024; Renz et al., 2025; Zhou et al., 2025) have arisen as a new paradigm, incorporating world knowledge from pre-trained VLMs to enhance performance in long-tail scenarios. Additionally, VLA architectures can accept human instructions to support human-preferred driving behaviors (Kim et al., 2024), while language serves as an interpretable intermediate representation for improved explainability (Tian et al., 2024; Wang et al., 2025).

**Beyond Imitation Learning.**    Current mainstream pipelines still operate within imitation learning-based frameworks, which suffer from causal confusion and lack verifiable safety guarantees. Many studies have attempted to address this issue, which can be broadly categorized as follows: 1) The model uses trajectory anchors, which are derived from clustered trajectory data or rule-based

proposals, as conditioning inputs and is designed to predict offsets for further trajectory refinement (Dauner et al., 2023). Hydra-MDP (Li et al., 2024) utilizes trajectory anchors as candidates for post-selection, while DiffusionDrive (Liao et al., 2024) employs anchors as starting points and uses a pseudo-diffusion process for refinement. Although these methods exhibit improved reliability, they rely heavily on rule-based design. 2) Reinforcement learning methods enhance model capabilities through exploration (Shalev-Shwartz et al., 2016; Kiran et al., 2021; Cao et al., 2023; Lu et al., 2023); for instance, GIGAFLOW (Cusumano-Towner et al., 2025) significantly improves performance via self-play in simulation. However, online rollouts are infeasible for real-world vehicle deployment, and simulation training faces the sim-to-real gap. Although recent advances in world models (Guan et al., 2024) offer a potential solution, they still struggle with out-of-distribution simulation. 3) Other methods, such as guidance mechanisms for diffusion models, enable the injection of reward signals during the denoising process (Jiang et al., 2023; Zhong et al., 2023). Diffusion Planner (Zheng et al., 2025) represents a pioneering effort in applying diffusion models to closed-loop planning tasks. Although it utilizes guidance to adjust behavior during inference, the method relies on additional gradient computations, resulting in high computational cost. In this paper, we propose a novel reflection mechanism based on discrete diffusion that naturally incorporates safety constraints through search, masking, and inpainting during trajectory generation.

## 3 PRELIMINARIES

### 3.1 AUTONOMOUS DRIVING PLANNING

We formulate the autonomous driving planning task as learning a conditional distribution $p(\tau \mid c)$, where the goal is to generate a future trajectory $\tau$. Each waypoint is expressed in the ego-vehicle frame, conditioned on a scene context $c$ that includes multi-view images, instructions, and ego-vehicle state. The primary challenge in planning is that trajectories must adhere to traffic rules and safety constraints, which is difficult for imitation learning-based methods due to the absence of explicit signals to ensure strict compliance with these requirements.

### 3.2 DISCRETE DIFFUSION

Discrete diffusion models (Austin et al., 2021; Meng et al., 2022; Lou et al., 2023) have emerged as a powerful non-autoregressive paradigm for generating structured sequences. This process is defined by a forward corruption process and a learned reverse denoising process.

**Forward and Reverse Process.** The forward process degrades a clean sequence of discrete tokens $\mathbf{y} = (\mathbf{y}_1, \dots, \mathbf{y}_i, \dots, \mathbf{y}_L)$ over a series of $S$ timesteps. At each step $s \in \{1, \dots, S\}$, a noisy version of the sequence, $\tilde{\mathbf{y}}^{(s)}$, is created by masking a subset of the tokens in $\mathbf{y}$. Specifically, a binary mask $\mathbf{m}^{(s)} = (m_1^{(s)}, \dots, m_i^{(s)}, \dots, m_L^{(s)}) \in \{0,1\}^L$ is sampled, and each token $\mathbf{y}_i$ is replaced with a special [MASK] token if $m_i^{(s)} = 1$. The number of masked tokens is determined by a noise schedule, such as a cosine schedule, which typically increases the masking ratio as $s$ approaches $S$. The core learning task is to train a model $p_\theta$ to reverse this corruption. This model learns to predict the original tokens at the masked positions, conditioned on the unmasked tokens, the timestep $s$, and any external context $c$. The model is trained by minimizing the negative log-likelihood objective:

$$\mathcal{L}(\theta) = \mathbb{E}_{\mathbf{y}, c, s, \mathbf{m}^{(s)}} \left[ - \sum_{i : m_i^{(s)} = 1} \log p_\theta\big(\mathbf{y}_i \mid \tilde{\mathbf{y}}^{(s)}, c, s\big) \right]. \tag{1}$$

Here, $s \in [0, 1]$ represents the masking ratio determined by the noise schedule, and $c$ encompasses the scene context including multi-view images, ego-status, and instructions.

**Model Inference.** To generate a new sequence, the process starts with a fully masked sequence, $\tilde{\mathbf{y}}^{(S)}$. The model then iteratively refines this sequence for $S$ steps. In each step, the model predicts a probability distribution for the tokens at the masked positions. A subset of these predictions is then sampled and fixed, while the rest are re-masked for the next refinement step. Specifically, we utilize a linear noise schedule. During inference, we adopt a parallel decoding strategy where, at

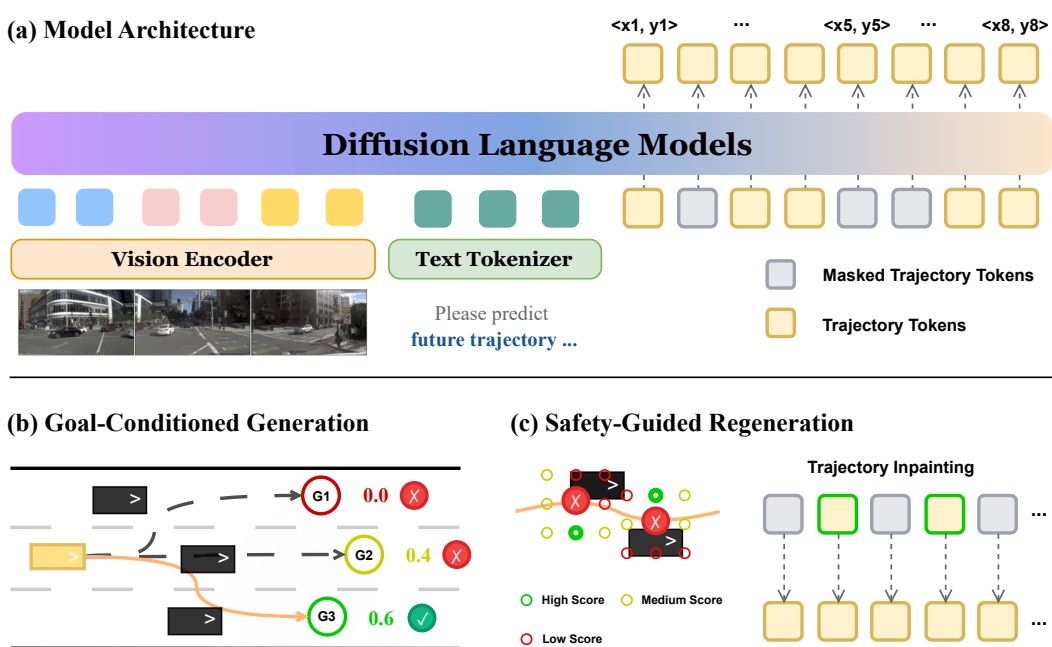

Figure 1: *ReflectDrive* Framework Overview.

each step, we select and fix a subset of tokens with the highest predicted confidence scores, allowing multiple tokens to be decoded simultaneously until the sequence is complete. A central advantage of this framework, and one especially critical to our work, is its capacity for inpainting, defined as the ability to reconstruct masked segments of a sequence while maintaining consistency with the context from unmasked tokens. Additionally, the discrete token structure supports efficient search and constraint integration, making it possible to guide trajectories using safety constraints.

## 4 METHOD

In this section, we present *ReflectDrive*, a novel learning-based framework that integrates a reflection mechanism to facilitate safe trajectory generation via discrete diffusion, as illustrated in Figure 1. We first introduce a trajectory discretization method tailored for integration into a masked diffusion process. A pre-trained diffusion language model is then employed for trajectory generation. Finally, we propose a reflection mechanism specifically designed to ensure safety during the trajectory generation process. This mechanism leverages diffusion inpainting and capitalizes on the advantages of discrete token spaces for efficient constraint-based search.

### 4.1 DISCRETE DIFFUSION FOR AUTONOMOUS DRIVING PLANNING

**Trajectory Discretization.** To represent continuous waypoints in a discrete format, we quantize each 2D coordinate $(x, y)$ by mapping its $x$ and $y$ values independently to the closest tokens in their respective 1D codebooks. We define a uniform 1D codebook $\mathcal{A} = \{a_1, a_2, \dots\}$ by discretizing a spatial range $[-M, M]$ with resolution $\Delta_g$. A quantizer $\mathcal{Q}$ maps a real value to its nearest token, and its inverse recovers the coordinate. Each 2D waypoint is thus represented by a token pair $(\mathbf{y}_{j,x}, \mathbf{y}_{j,y})$, and the full trajectory becomes a flattened sequence $\mathbf{y} = \mathcal{Q}(\tau) = (\mathbf{y}_{1,x}, \mathbf{y}_{1,y}, \dots, \mathbf{y}_{N,x}, \mathbf{y}_{N,y}) \in \mathcal{A}^{2N}$. At first glance, discretization may appear to cause some loss in trajectory precision. However, in practical deployment, the resolution can be adjusted to control accuracy, or different codebook partitioning strategies can be employed. Specifically, we utilize a grid resolution of $\Delta g = 0.3$ meters. Given the spatial range of $[-100, 100]$ meters, this results in a codebook size of $|A| \approx 667$ tokens per dimension. Most importantly, discretization facilitates efficient search for feasible solutions in the Bird's-Eye View (BEV) space. Experimental results in Section 5.2 and Figure 3 further

demonstrate that, with discrete representations, our reflection mechanism significantly enhances the safety of the generated trajectories.

**Discrete Diffusion Model.**    Based on our discretized trajectory representation, we instantiate the trajectory planner using the discrete diffusion framework described in Section 3. In practice, we employ a VLA model as the planner, initialized from a pre-trained Diffusion Language Model (You et al., 2025; Nie et al., 2025) that exhibits strong pre-training performance in understanding driving scenarios. The model can generate a tokenized trajectory $\mathbf{y}$ conditioned on a scene context $c$ (multi-view images, language instruction, ego state). The model is trained via the denoising objective in Eq. 1 using autonomous driving planning datasets for supervised fine-tuning. This provides the inherent capability for bidirectional inpainting, which serves as the foundation of our method. It enables the model to perform holistic parallel refinement and elegantly repair trajectories around externally guided safety edits during the reflective inference process.

## 4.2    REFLECTIVE INFERENCE

With the discrete diffusion-based VLA model as our foundation, we introduce a reflective inference framework to bridge the gap between imitation learning and safety-critical deployment. This framework operates in two stages: goal-conditioned trajectory generation and safety-guided regeneration. The entire process is guided by a set of specialized scoring functions.

**Scoring Function Definitions.**    To systematically evaluate trajectories, our framework incorporates three distinct scoring functions. The detailed composition of these functions, which are designed based on established autonomous driving evaluation principles, is provided in Appendix E.

- *Global Scorer ($S_{global}(\tau)$):* This scorer evaluates the overall quality of a complete trajectory, considering both safety and coherence, and returns a value of zero if any critical rule is violated.

- *Safety Scorer ($S_{safe}(\tau)$):* This scorer acts as a safety oracle to identify specific points of failure.

- *Local Scorer ($S_{local}(a_x, a_y)$):* This scorer evaluates each candidate token pair $(a_x, a_y)$ using a comprehensive function that assesses its impact on the trajectory's safety and coherence.

**Goal-Conditioned Generation.**    To ensure our planner can reason about high-level, global intents that go beyond simple local adjustments, the process begins with generating a diverse set of trajectory proposals. This procedure is essential for multi-modal driving behavior modeling and serves as a necessary step for subsequent regeneration. Since the local search in our safety-aware regeneration stage is intentionally constrained for efficiency, it cannot accommodate large-scale changes, such as taking a different turn at an intersection, which require broader exploration. We first use the model to produce a probability distribution for the terminal waypoint tokens, $p_\theta(\mathbf{y}_N \mid c, s)$, where $\mathbf{y}_N = (\mathbf{y}_{N,x}, \mathbf{y}_{N,y})$. From this distribution, we sample a set of high-probability goal candidates. We then apply Non-Maximum Suppression (NMS) (Ren et al., 2015) to obtain a spatially diverse set of $K$ candidate goals, $\mathcal{G} = \{G_1, \ldots, G_K\}$:

$$\mathcal{G} = \text{NMS}\left(\text{TopK}_{K'}\left(p_\theta(\mathbf{y}_N \mid c, s)\right), d_{\text{NMS}}, K\right) \tag{2}$$

where $\text{TopK}_{K'}(\cdot)$ is an operator that selects the $K'$ most probable goal candidates from the model's output distribution. The $\text{NMS}(\cdot)$ function then filters this set using a distance threshold $d_{\text{NMS}}$ to produce the final, spatially diverse set $\mathcal{G}$ of size $K$. For practical deployment, a dedicated goal generation model could be used to improve the accuracy and quality of goal points. However, for simplicity, we employ the same model for both goal generation and trajectory planning. Then, for each goal $G_k \in \mathcal{G}$, we generate a full trajectory $\tau_k$ by sampling from the conditional distribution $p_\theta(\mathbf{y}_{1:2N-2} \mid G_k, c, s)$ via inpainting. The resulting $K$ trajectories are evaluated using the **Global Scorer** $S_{\text{global}}(\cdot)$, which assesses each plan based on a combination of metrics including goal progress. The top-scoring trajectory $\tau^*$ is then selected for further refinement.

$$\tau^* = \underset{\tau_k, k=1, \ldots, K}{\arg\max} S(\tau_k). \tag{3}$$

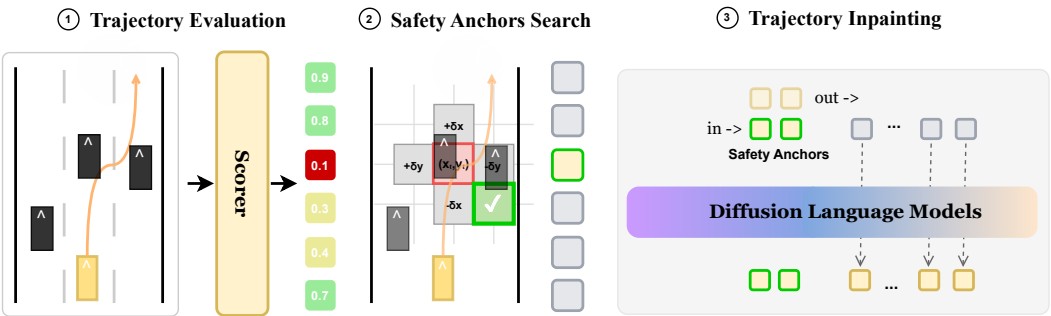

Figure 2: Safety-Guided Regeneration Pipeline.

**Safety-Guided Regeneration.** The selected trajectory $\tau^*$, while coherent, may still violate physical constraints. We address this with an iterative, gradient-free refinement loop that forms a dialogue between the generative model and an external safety oracle, as shown in Figure 2.

- *Trajectory Evaluation.* The process begins when the **Safety Scorer** $S_{\text{safe}}(\cdot)$ evaluates the dequantized trajectory and identifies the specific waypoints that are unsafe. The oracle assigns a safety score to each original waypoint based on the worst violation (e.g., drivable area infraction) within a local time window. This allows it to precisely pinpoint unsafe waypoints.

- *Safety Anchors Search.* For the earliest waypoint that violates a safety threshold, we perform a highly efficient local search within a small Manhattan neighborhood $\mathcal{N}_\delta$ of the original tokens to identify an improved token pair, rather than resorting to complex continuous optimization. The corrected token pair that maximizes the local safety score is then designated as a safety anchor.

- *Trajectory Inpainting.* We then leverage the diffusion model's powerful inpainting capability to regenerate the surrounding trajectory segments conditioned on safety anchors. This single-pass regeneration allows the model to naturally re-establish global coherence around the safety-driven edit. This cycle of identifying violations, performing discrete corrections, and re-inpainting continues until the plan is fully safe or a computational budget is met. Specifically, we set a maximum budget of 10 iterations to ensure real-time feasibility. If the algorithm reaches this limit without finding a fully safe trajectory, it outputs the candidate with the highest safety score found during the search as a fallback strategy.

This refinement process operates as an iterative loop. In each iteration, the top-scoring trajectory $\tau^*$ is evaluated by the **Safety Scorer** to identify the set of all violating waypoint indices $\mathcal{V} = \{t \mid S_{\text{safe}}(\tau^*)_t < \tau_{\text{safe}}\}$. If $\mathcal{V}$ is empty, the trajectory is deemed safe and the process terminates. Otherwise, we select a subset $\mathcal{T}^* \subseteq \mathcal{V}$ corresponding to the segment of violations. For each timestep $t \in \mathcal{T}^*$, the **Local Scorer** is employed to find an improved token pair within a local neighborhood $\mathcal{N}_\delta$ by solving:

$$(\mathbf{y}'_{t,x}, \mathbf{y}'_{t,y}) = \underset{(a_x, a_y) \in \mathcal{N}_\delta(\mathbf{y}_{t,x}, \mathbf{y}_{t,y})}{\arg\max} S_{\text{local}}(a_x, a_y). \tag{4}$$

The original tokens at the indices in $\mathcal{T}^*$ are replaced by these optimized pairs, which serve as fixed safety anchors for the subsequent inpainting step. The refinement cycle then continues with this updated trajectory. In practice, the reflective inference process is designed for real-time performance. The local search for corrective tokens is efficient, as it operates over a small, discrete neighborhood (e.g., a Manhattan distance $\delta \leq 10$) rather than requiring expensive gradient-based optimization. In practice, we find that most safety violations are resolved within 1–3 iterations of reflection, resulting in a manageable inference overhead.

## 5 EXPERIMENTS

### 5.1 BENCHMARK AND BASELINES

**Evaluation Setups.** In our implementation, the VLA model backbone is initialized from a publicly available pre-trained Vision-Language Model (LLaDA-V You et al. (2025)) and utilizes classifier-free guidance for trajectory generation. Input images are obtained from the front, front-left, and

Table 1: **NAVSIM Closed-Loop Results.** Methods are grouped by their core architectural paradigm. The † symbol denotes our method using a privileged ground-truth oracle for reflection, serving as an analytical upper bound. Best result per column is in **bold** (higher is better).

| Method | Paradigm | Input | NC↑ | DAC↑ | TTC↑ | Comf.↑ | EP↑ | PDMS↑ |
|---|---|---|---|---|---|---|---|---|
| *Base End-to-End Planners* | | | | | | | | |
| UniAD | - | Cam | 97.8 | 91.9 | 92.9 | **100.0** | 78.8 | 83.4 |
| PARA-Drive | - | Cam | 97.9 | 92.4 | 93.0 | 99.8 | 79.3 | 84.0 |
| Transfuser | - | C & L | 97.7 | 92.8 | 92.8 | **100.0** | 79.2 | 84.0 |
| *Augmented End-to-End Planners* | | | | | | | | |
| Hydra-MDP | - | C & L | 98.3 | 96.0 | 94.6 | **100.0** | 78.7 | 86.5 |
| DiffusionDrive | Diffusion | C & L | 98.2 | 96.2 | 94.7 | **100.0** | 82.2 | 88.1 |
| GoalFlow | Diffusion | C & L | 98.4 | 98.3 | 94.6 | **100.0** | 85.0 | 90.3 |
| *VLA Planners* | | | | | | | | |
| AutoVLA (Post-RFT) | Autoregressive | Cam | 98.4 | 95.6 | 98.0 | 99.9 | 81.9 | 89.1 |
| ReflectDrive (w/o R.I.) | Discrete Diffusion | Cam | 96.9 | 95.4 | 92.2 | **100.0** | 79.0 | 84.8 |
| ReflectDrive (Ours) | Discrete Diffusion | Cam | 97.7 | 99.3 | 93.5 | **100.0** | 86.9 | 91.1 |
| ReflectDrive† | Discrete Diffusion | Cam | **99.7** | **99.5** | **99.1** | 99.9 | **88.9** | **94.7** |
| *Human* | – | – | 100.0 | 100.0 | 100.0 | 99.9 | 87.5 | 94.8 |

front-right cameras. The language instruction provides a high-level navigational command, such as "turn left" or "go straight," along with textual descriptions of the ego vehicle's status. We evaluate our model on the large-scale real-world autonomous driving benchmark NAVSIM (Dauner et al., 2024) for closed-loop performance assessment. Following the official protocol, performance is reported with the PDMS score (higher is better), aggregated from five metrics: *NC* (no-collision rate), *DAC* (drivable area compliance), *TTC* (time-to-collision safety), *Comfort* (bounded acceleration/jerk) and *EP* (ego progress). We run all the methods under the official closed-loop simulator and report averages on the public test split. Our planner uses camera-only inputs unless otherwise stated; we also include Camera+LiDAR baselines to provide a more comprehensive comparison.

**Baselines.** We compare *ReflectDrive* to other autonomous driving systems. For example, vanilla E2E planners that purely use sensor information as input and output trajectories, such as UniAD (Hu et al., 2023), Para-Drive (Weng et al., 2024), Transfuser (Chitta et al., 2023). As well as augmented E2E planners that incorporate clustering results as auxiliary information like Hydra-MDP (Li et al., 2024), DiffusionDrive (Liao et al., 2024), and GoalFlow (Xing et al., 2025), the PDMS scores will be higher than vanilla E2E planners due to additional information. We also include recent AutoVLA (Zhou et al., 2025) model that unifies reasoning and action generation within a single autoregressive generation model, the PMDS score is the highest among VLA planners. For our model family, the table lists: *ReflectDrive (w/o R.I.)* trained with discrete masked diffusion adding classifier-free guidance at inference without reflective inference; *ReflectDrive* adding goal-conditioned generation and safety-guided regeneration, where the safety-guided regeneration relies on the reward model where surrounding obstacles are moving at constant speeds; *ReflectDrive†* adding goal-conditioned generation and safety-guided regeneration, where the safety-guided regeneration relies on the reward model where surrounding obstacles are ground-truth agents.

## 5.2 Main Results

Evaluation results on the NAVSIM benchmark are presented in Table 1.

**Base Model Validation.** *ReflectDrive* base model achieves the PDMS score 84.8 comparable to the base end-to-end models, such as UniAD, PARA-Drive, and Hydra-MDP, and slightly lower than the score of Augmented End-to-End Planners. However, it has not yet demonstrated significant performance advantages. We identify two potential limiting factors: first, the limited scale of training data, and second, room for improvement in the base VLM model's capabilities.

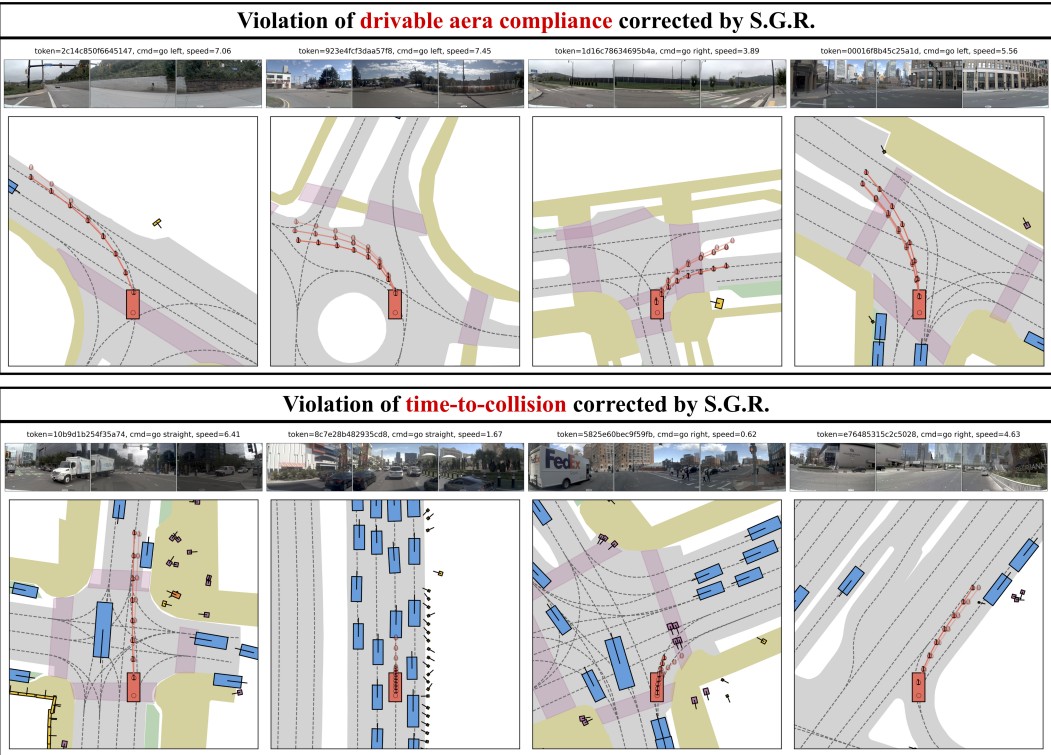

Figure 3: **Safety-Guided Regeneration (S.G.R) Visualization.** The first row illustrates three scenarios where large-angle turns are prone to boundary violations. The initial trajectories (lightest color) carry the risk of exceeding the boundaries. Using S.G.R, the trajectory is gradually optimized toward the safe region (with its color darkening progressively), ultimately resulting in a feasible trajectory. The second row depicts three scenarios involving intense interactions. Initial trajectories may pose collision risks with other vehicles or pedestrians. Through the iterative optimization of S.G.R., the trajectories learn to avoid conflicts or decelerate to yield, achieving much higher safety.

**Significant Improvements from Reflective Inference.** The introduction of safety-guided regeneration mechanism yields substantial improvements in safety metrics such as *DAC*, *TTC* and *NC*. This is primarily due to our reward function design that fully considers safety-related factors. For *EP* metrics, we employ a goal-conditioned generation strategy for optimization. Compared to *Reflect-Drive (w/o R.I.)*, *DAC* gets **+3.9-point** improvement,*TTC* gets **+1.3-point** improvement, *NC* gets **+0.8-point** improvement and *EP* gets **+7.9-point** improvement. while ensuring trajectory safety without compromising progress. Compared to other end-to-end planners, *DAC* significantly outperforms others and approaches human-level performance, while *TTC* and *NC* underperform expectations due to the use of constant-velocity agents, which can lead to inaccurate safety estimations in safety-critical scenarios.To explore the upper bound of *ReflectDrive*, we therefore employ ground-truth agent states in our evaluation.

**Approaching Human Driving Performance.** When using ground truth agents information (i.e., with complete environmental information), the performance of the system already matches human driving trajectories, such as *NC* **99.7**, *DAC* **99.5**, *TTC* **99.1**, even *EP* **88.9** which is higher than human to demonstrate the potential powerful capabilities of *ReflectDrive*. Compared to ReflectDrive based on constant velocity agents, *DAC* gets **+0.2-point** improvement, *TTC* gets **+5.6-point** improvement, *NC* gets **+2.0-point** improvement and *EP* gets **+2.0-point** improvement, which meet the expectations. This implies that further performance improvements can be achieved with more accurate detection and prediction results—a concern that is mitigated in practical deployment, as specialized models are dedicated to these tasks. And through failure case analysis in Figure 6, we identified optimization opportunities in the search algorithm. With further optimization of the search algorithm, we expect to comprehensively surpass human driving performance.

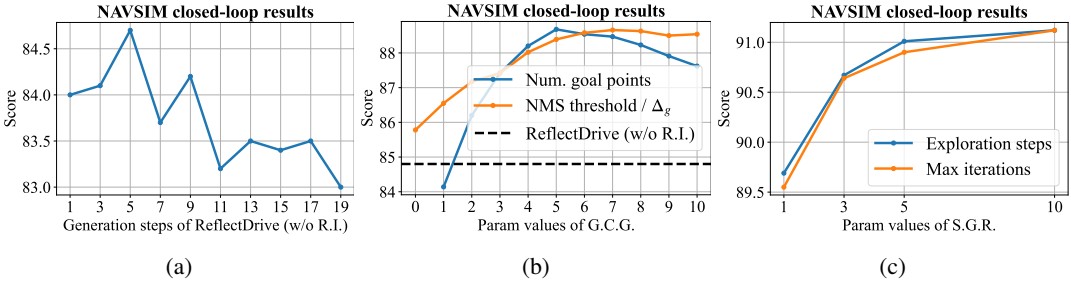

Figure 4: Ablation on (a) the number of generation steps for *ReflectDrive (w/o R.I.)*, (b) the number of goal points and range of NMS for Goal-Conditioned Generation (G.C.G.), and (c) the numbers of exploration steps as well as max iterations for Safety-Guided Regeneration (S.G.R.).

Table 2: **Ablation for Reflective Inference.** The ablation study results of goal-conditioned generation and safety-guided regeneration to demonstrate the effectiveness of reflective inference.

| Method | Goal-Cond. | Safety-Guided | NC↑ | DAC↑ | TTC↑ | Comf.↑ | EP.↑ | PDMS↑ |
|---|---|---|---|---|---|---|---|---|
| W/o Both | × | × | 96.9 | 95.4 | 92.2 | 100.0 | 79.0 | 84.8 |
| W/ Goal-Cond. | ✓ | × | 96.6 | 96.5 | 91.5 | 100.0 | 83.8 | 87.4 |
| W/ Safety-Guided | × | ✓ | 98.1 | 98.9 | 94.8 | 99.9 | 84.1 | 90.3 |
| **Full Model** | ✓ | ✓ | 97.7 | 99.3 | 93.5 | 99.9 | 86.9 | 91.1 |

## 5.3 QUALITATIVE RESULTS

To further demonstrate the capabilities of *ReflectDrive*, we show the trajectory generation results of representative scenarios, as shown in Figure 3. *ReflectDrive* shows high-security trajectory generation, where the initial trajectory has the risk of going out of bounds, but with reflective inference as guidance, the trajectory gradually iterates and optimizes toward the safe region, ultimately producing a feasible trajectory. It is noteworthy that the generated trajectories remain kinematically feasible and smooth even after discretization, further demonstrating the viability of using discrete diffusion for autonomous driving planning. We also provide additional good examples in Figure 5.

## 5.4 ABLATION STUDIES

**Ablation on Inference Parameters.** We conducted ablation experiments on key adjustable parameters involved in the generation and reflection process, with results presented in Figure 4. These parameters include: **Generation steps**, which governs the number of steps for impainting trajectories in our discrete diffusion model; **Num. goal points**, indicating the number of selected goal points (i.e., the number of multi-modal candidates); **Exploration steps**, controlling the search range for candidate points (with larger values providing more correction space); and **Max iterations**, denoting the maximum number of regeneration iterations. For diffusion generation steps, the results reveal a non-monotonic relationship between performance and the number of steps: model performance improves during the initial steps, peaks at 5 steps, and subsequently declines with additional steps. Furthermore, we demonstrate that multi-modal behavior modeling can further improve model performance and offer a wider range of options for selection. Lastly, we observe the presence of inference scaling: as computational resources allocated to exploration and regeneration steps increase, model inference performance improves accordingly. The upper bound of this scaling may also depend on the strategy employed, indicating potential for further optimization in future work.

**Design Choices for Reflective Inference.** Based on the optimal parameter configuration, we conducted ablation experiments on goal-conditioned generation and safety-guided regeneration methods. As shown in Table 2, the results indicate that goal-conditioned generation enhances ego progress, while safety-guided regeneration improves both safety metrics and progress performance. These findings validate the complementary nature of our *ReflectDrive* approach, where goal-conditioned generation focuses on progress optimization while safety-guided regeneration ensures safety constraints are met without compromising driving efficiency.

## 6 CONCLUSION

We propose *ReflectDrive*, a novel learning-based framework that integrates a reflection mechanism for safe trajectory generation via discrete diffusion. The two-dimensional driving space is discretized into an action codebook, enabling fine-tuning of pre-trained Diffusion Language Models for planning tasks. Our reflection mechanism begins with goal-conditioned generation to capture diverse multi-modal behaviors, followed by safety-guided regeneration that identifies feasible solutions through gradient-free inpainting. Evaluations on the NAVSIM benchmark demonstrate the effectiveness and safety advantages of our approach. Due to space limitations, further discussions on limitations and future directions are provided in Appendix G.

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

APPENDIX

## A    VISUALIZATION OF PLANNING RESULTS

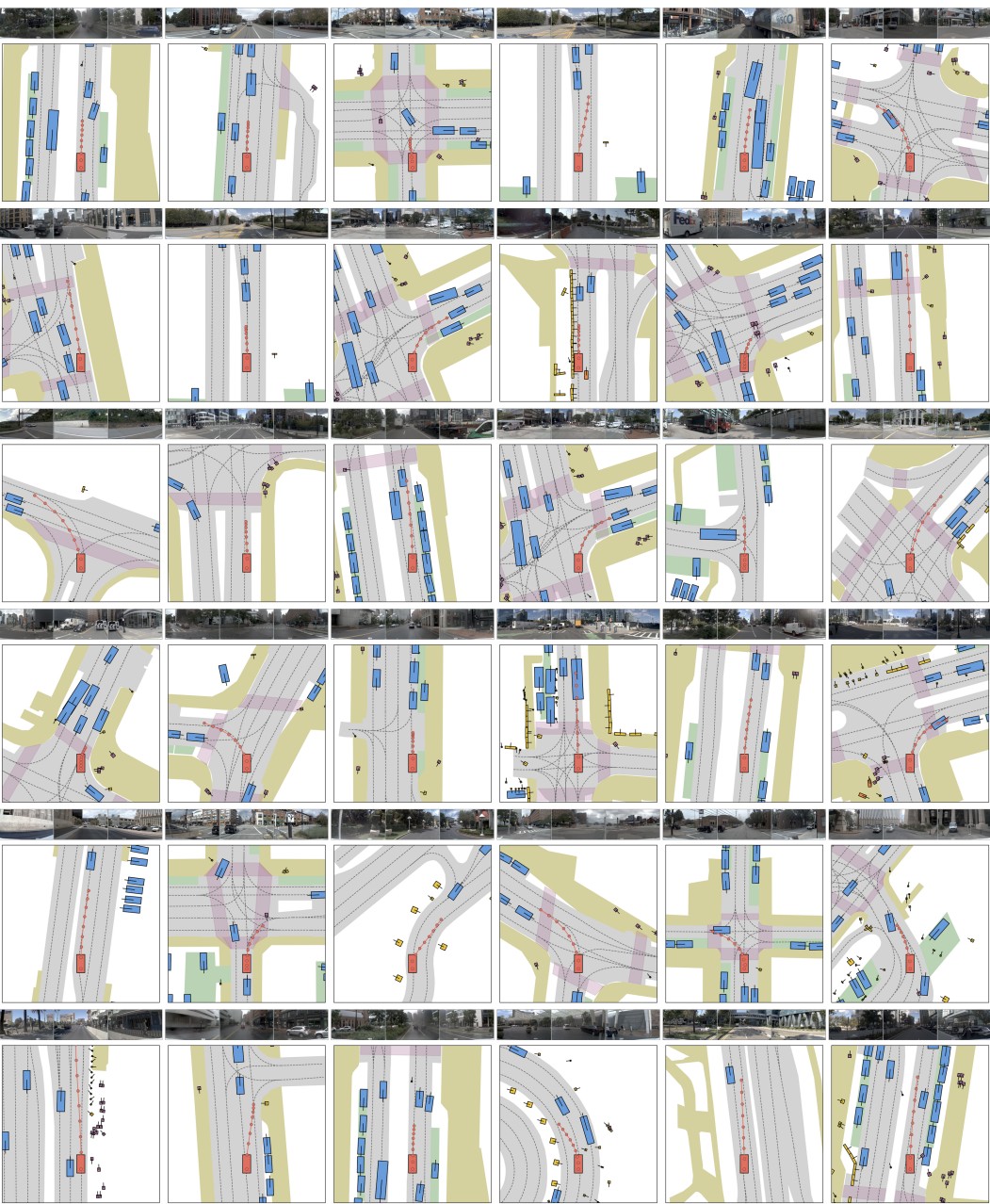

Figure 5: Planning results that meet the PDM evaluation criteria.

## B    SUPERVISED FINE-TUNING (SFT) DETAILS

Table 3 shows the parameters used in our inference stage. We fixed the length of the output because the number of trajectory points is always the same, and we perform parallel decoding for all trajectory points. We generate 3 diverse goal proposals to ensure good coverage of potential driving intents. A threshold of 0.9 meters is used during non-maximum suppression to ensure that the

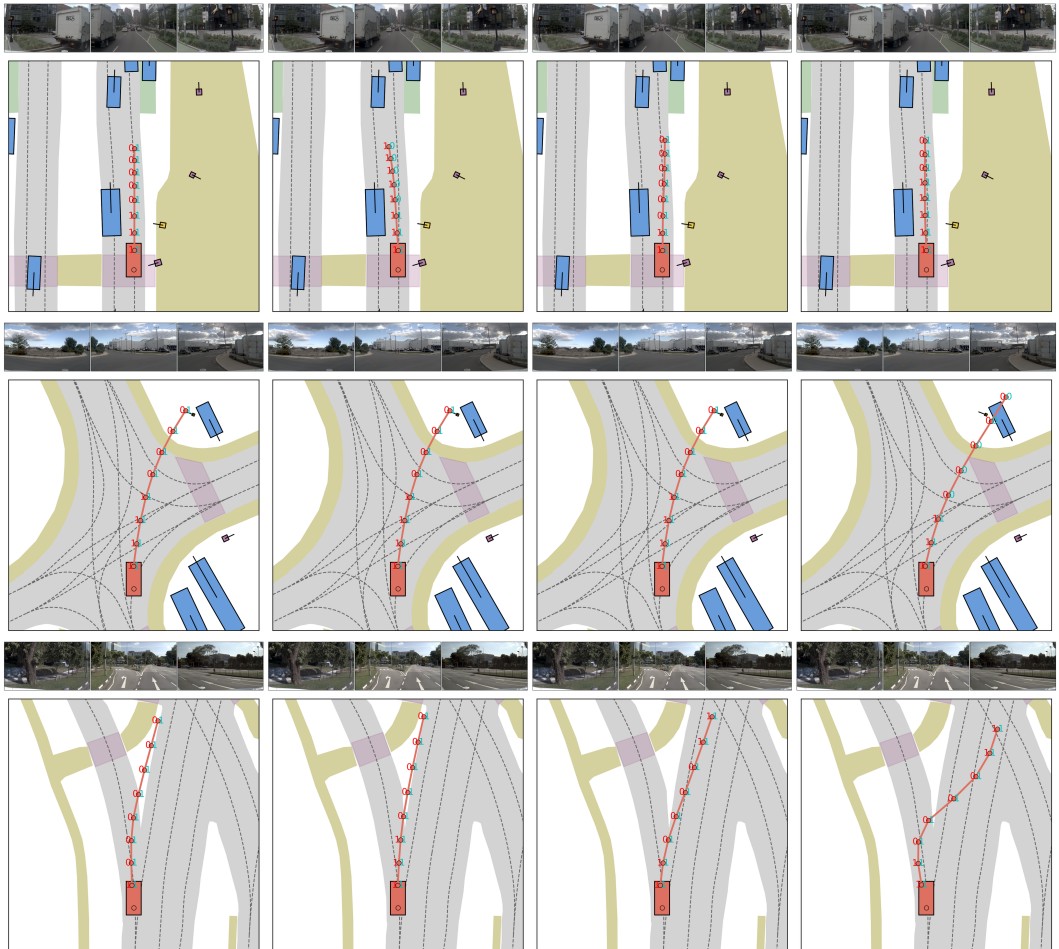

Figure 6: Planning results of bad cases. Row 1 shows the oscillation between boundaries and needs to improve the reward, such as adding the distance from the centerline in the future. Row 2 shows goal point selection deviation. Row 3 shows navigation deviation.

selected goal points are spatially distinct. The safety loop is capped at 10 iterations to guarantee a fixed upper bound on inference time. In practice, most trajectories converge to a safe state in 1-3 iterations. Tab 4 shows the key hyperparameters in our training stage. Specifically, our model was initialized from the pre-trained LLaDA-V checkpoint and fine-tuned on the navtrain split of NAVSIM, which contains 80k annotated samples.

Table 3: Inference Configuration for ReflectDrive.

| Parameter | Value |
|---|---|
| Steps | 5 |
| Answer length | 32 |
| Block length | 32 |
| Remask | low-confidence |
| Number of goal candidates ($K$) | 3 |
| NMS distance threshold ($d_{\mathrm{NMS}}$) | 0.9 |
| Max refinement iterations | 10 |

Table 4: Key Hyperparameters for Training

| Parameter | Value |
|---|---|
| Spatial Range ($M$) | [-100, 100] |
| Batch Size | 16 |
| Gradient Accumulation Steps | 1 |
| Learning Rate | $1 \times 10^{-5}$ |
| Training Epochs | 3 |
| Max Context Length | 8192 |
| Learning Rate Scheduler | Cosine |
| Warmup Ratio | 0.03 |
| Weight Decay | 0.0 |
| Precision | bfloat16 |

## C  INFERENCE LATENCY AND SYSTEM PROFILE

We evaluate the latency and memory profile using a single NVIDIA H20 GPU. Our current implementation is a research prototype based on the LLaDA-V backbone without engineering optimizations such as KV-caching, quantization, or C++ deployment. Table 5 details the inference time breakdown.

Table 5: Inference Latency Breakdown (Single NVIDIA H20).

| Component | Time (s) | Note |
|---|---|---|
| **Stage 1: Goal-Conditioned Gen.** | **6.82** | Total |
| - Goal Proposal | 0.62 | Generation |
| - Trajectory Inpainting | 6.06 | - |
| - Scoring & Selection | 0.15 | - |
| **Stage 2: Safety-Guided Regen.** | **4.88** | Per Iteration |
| - Safety Check | 0.13 | - |
| - Search | 0.84 | - |
| - Scoring & Selection | 1.68 | - |
| - Regeneration (Inpainting) | 2.14 | - |
| **Average Total Inference** | **~8.92** | - |

The latency is currently dominated by the VLM backbone's forward pass and the Python-based search logic. The search overhead (2.52s) can be reduced to milliseconds through C++ optimization, and model inference can be accelerated using standard techniques such as KV caching. In the worst-case scenario (max 10 iterations), the latency would be higher, but empirical results show that safety violations are resolved within 0.43 iterations on average.

## D  ADDITIONAL ABLATION STUDIES

### D.1  DISCRETIZATION GRANULARITY

We investigate the impact of grid resolution $\Delta g$ on performance. As shown in Table 6, using an excessively fine granularity ($\Delta g = 0.1$m) leads to a significant performance drop (88.2). This is likely due to the drastically increased codebook size ($|A| = 2000$), which complicates the classification task for the diffusion model. Conversely, the model exhibits robust high performance ($> 90.7$) across the range of 0.2m to 0.5m. We selected $\Delta g = 0.3$m as the default because it achieves performance comparable to the peak while maintaining a significantly smaller vocabulary size, offering a better balance between precision and model complexity.

Table 6: Ablation on Discretization Granularity ($\Delta g$).

| Resolution $\Delta g$ (m) | Codebook Size $|A|$ | PDMS |
|:---:|:---:|:---:|
| 0.1 | 2000 | 88.2 |
| 0.2 | 1000 | 91.3 |
| 0.3 (Default) | 667 | 91.1 |
| 0.4 | 500 | 90.7 |
| 0.5 | 400 | 91.2 |

## D.2 ROBUSTNESS OF SCORING FUNCTIONS

To address concerns about reward shaping, we conducted a sensitivity analysis on the scorer weights ($w_{EP}, w_{TTC}, w_C$) and the binary TTC threshold. As shown in Table 7, our method demonstrates strong robustness to hyperparameter variations. Removing specific weights (e.g., $w_{EP} = 0$) results in minimal performance fluctuation, confirming that the gains stem from the reflection mechanism rather than overfitting to metric weights.

Table 7: Sensitivity Analysis of Scorer Weights and TTC Threshold.

| $w_{EP}$ | $w_{TTC}$ | $w_C$ | TTC Thresh (s) | PDMS |
|:---:|:---:|:---:|:---:|:---:|
| **5 (Default)** | 5 | 2 | **1.0** | **91.1** |
| 5 | 5 | 0 | 1.0 | 91.1 |
| 2 | 5 | 5 | 1.0 | 91.1 |
| 0 | 5 | 5 | 1.0 | 91.2 |
| 5 | 2 | 5 | 1.0 | 91.1 |
| 5 | 0 | 5 | 1.0 | 90.9 |
| 5 | 5 | 2 | 0.5 | 90.0 |
| 5 | 5 | 2 | 1.5 | 91.0 |

## D.3 SENSITIVITY OF GOAL PROPOSAL PARAMETERS

In practice, we selected a smaller number of goals ($K = 3$) from an efficiency perspective, requiring them to maintain a certain distance, and chose $d_{NMS}$ through visualization, rather than based on actual scores. Table 8 confirms that $K = 3$ captures the majority of the performance gains (+3.3 PDMS over $K = 1$) while maintaining computational efficiency compared to higher values like $K = 5$.

We further illustrate this choice through qualitative analysis. Figure 7 demonstrates that relying solely on the Top-1 candidate ($K = 1$, shown in **Black**) often leads to failure cases, whereas increasing $K$ to 3 introduces valid alternatives (Top-2 and Top-3, shown in **Red**). Additionally, Figure 8 visualizes the effect of the NMS threshold. While a larger threshold (2.1m) increases spatial diversity compared to our default (0.9m), it introduces candidates with higher variance that can be challenging for the subsequent refinement stage, reinforcing our selection of 0.9m as a robust baseline.

## E SCORING FUNCTION IMPLEMENTATION DETAILS

This appendix provides the detailed composition of the scoring functions introduced in the main text. Our evaluation framework is designed to be comprehensive, balancing hard safety constraints with continuous measures of driving quality and efficiency.

The final score for a trajectory, which underpins our $S_{global}$ and $S_{local}$ scorers, is computed as a product of a Hard Safety Compliance term ($H(\tau)$) and a Performance Quality term ($Q(\tau)$).

Table 8: Ablation on Goal Proposal Parameters (GCG Stage Only).

**(a) Goal Points ($K$)**

| $K$ | 1 | 3 (Def) | 5 | 10 |
|---|---|---|---|---|
| PDMS | 84.1 | **87.4** | 88.7 | 87.6 |

**(b) NMS Threshold ($d_{\text{NMS}}$)**

| Thresh. (m) | 0.0 | 0.3 | 0.9 (Def) | 1.2 |
|---|---|---|---|---|
| PDMS | 85.8 | 86.5 | **87.4** | 88.0 |

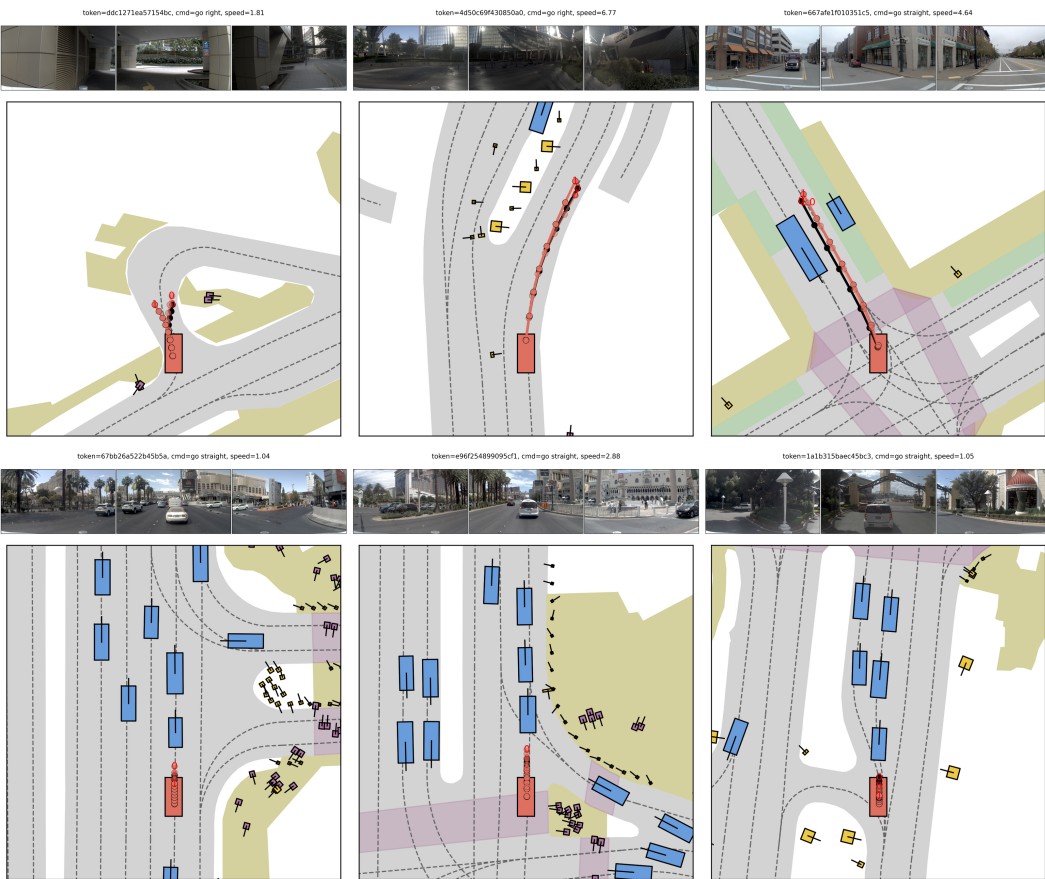

Figure 7: **Qualitative Visualization of Goal Candidates ($K = 1$ vs. $K = 3$).** The **Black** trajectories represent the Top-1 choice (equivalent to $K = 1$), which fails in these challenging scenarios. The **Red** trajectories represent the Top-2 and Top-3 candidates introduced by setting $K = 3$. In these cases, the alternative red trajectories successfully avoid obstacles or boundaries, demonstrating how multimodal proposals improve robustness.

### E.1 HARD SAFETY COMPLIANCE TERM ($H(\tau)$)

This term acts as a safety gatekeeper. It is the product of several individual metric scores, each corresponding to an inviolable driving rule. If any rule is broken, this entire term approaches zero, effectively nullifying the trajectory's score regardless of its performance quality.

$$H(\tau) = m_{\text{NC}}(\tau) \cdot m_{\text{DAC}}(\tau) \qquad (5)$$

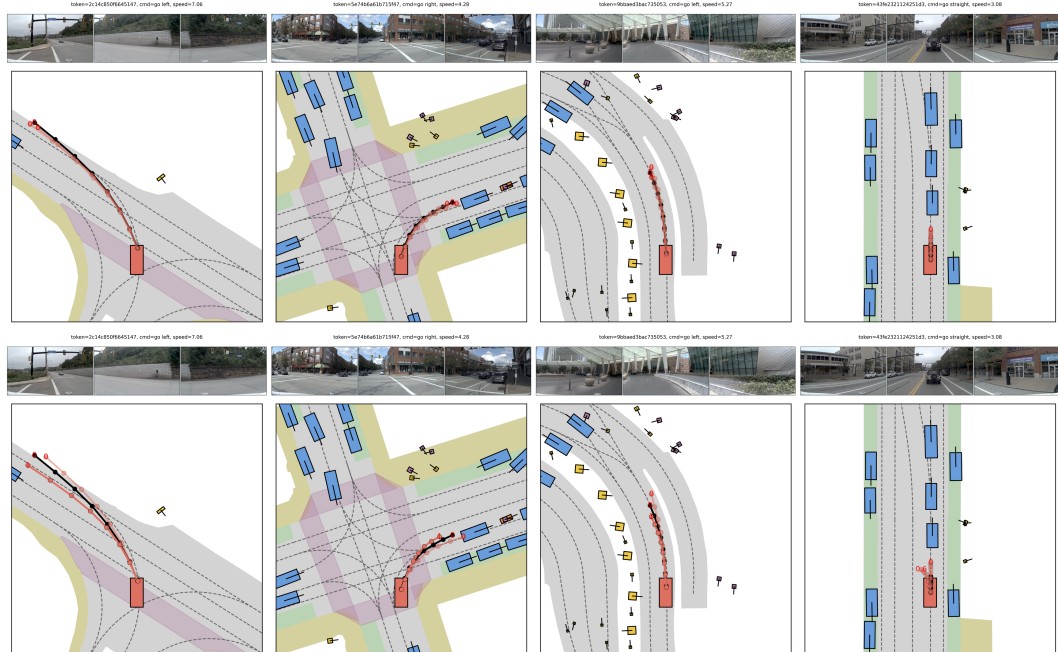

Figure 8: **Visualization of NMS Threshold Effects ($d_{\text{NMS}} = 0.9$m vs. $2.1$m).** The **top row** displays candidates generated with the default threshold (0.9m), exhibiting tighter clustering. The **bottom row** shows the same scenarios with a larger threshold (2.1m), which forces greater spatial separation between candidates.

The individual metrics are defined as follows:

- $m_{\text{NC}}$ **(No at-fault Collision):** This metric penalizes collisions for which the ego vehicle is deemed responsible. A collision is considered "at-fault" if the ego vehicle's front collides with any object, or if it collides with a static object.
  - Score = 1.0: No at-fault collision occurs.
  - Score = 0.5: An at-fault collision with a static object occurs.
  - Score = 0.0: Any other at-fault collision occurs.
- $m_{\text{DAC}}$ **(Drivable Area Compliance):** This is a strict binary metric that ensures the vehicle remains within the legally designated drivable area.
  - Score = 1.0: The vehicle's entire footprint remains within the drivable area.
  - Score = 0.0: Any part of the vehicle's footprint goes outside the drivable area.

Our *Safety Scorer ($S_{safe}$)* uses this exact logic, evaluating these hard constraints at each waypoint to detect failures.

### E.2 PERFORMANCE QUALITY TERM ($Q(\tau)$)

This term evaluates the quality of a trajectory that has passed the hard safety checks. It is a normalized weighted sum of several performance metrics.

$$Q(\tau) = \frac{w_{\text{EP}} \cdot m_{\text{EP}}(\tau) + w_{\text{TTC}} \cdot m_{\text{TTC}}(\tau) + w_{\text{C}} \cdot m_{\text{C}}(\tau)}{w_{\text{EP}} + w_{\text{TTC}} + w_{\text{C}}} \tag{6}$$

The individual metrics and their weights are as follows:

- $m_{\text{EP}}$ **(Ego Progress):** This metric measures the vehicle's progress along its intended high-level route. The value is normalized to a range of [0, 1] based on a feasible upper bound for progress in the given scene.

    – Weight ($w_{\mathrm{EP}}$): 5

- $m_{\mathrm{TTC}}$ **(Time-to-Collision):** This metric ensures a safe temporal buffer to other agents. It is a binary score based on a predefined safety threshold.
    - Score = 1.0: The minimum TTC to any other agent remains above the safe threshold (e.g., 2.0 seconds).
    - Score = 0.0: The minimum TTC drops below the threshold.
    - Weight ($w_{\mathrm{TTC}}$): 5

- $m_{\mathrm{C}}$ **(Comfort):** This metric evaluates ride smoothness. It is a binary score based on whether the vehicle's dynamics stay within acceptable bounds.
    - Score = 1.0: Longitudinal and lateral acceleration and jerk all remain within predefined comfort limits.
    - Score = 0.0: Any of the dynamic limits are exceeded.
    - Weight ($w_{\mathrm{C}}$): 2

# F  SCALABILITY AND COMPARISON WITH CONTINUOUS DIFFUSION

To further validate the effectiveness and scalability of our discrete diffusion framework, we conducted a large-scale open-loop evaluation using an in-house dataset comprising approximately 1 billion samples. This dataset offers comprehensive coverage of diverse and complex driving scenarios, far exceeding the scale of public benchmarks.

We compared our discrete diffusion VLA against a continuous diffusion VLA baseline trained on the same data. Both models generated 8 trajectories per sample. We report the Average Displacement Error (ADE) and Final Displacement Error (FDE) at 40m, 80m, and 120m horizons. *Top1* refers to the trajectory with the highest predicted confidence, while *Min* refers to the best trajectory among the 8 samples (oracle selection).

As shown in Table 9, the discrete diffusion VLA consistently outperforms the continuous diffusion baseline across most metrics, particularly in long-horizon prediction (120m FDE: 2.19 vs. 2.71). This empirical evidence suggests that discrete tokenization effectively captures complex multi-modal distributions at scale, validating it as a robust alternative to continuous parameterization for autonomous driving planning.

Table 9: Open-Loop Comparison on Large-Scale In-House Dataset (1B Samples).

| Method | Top1 FDE | | | Top1 ADE | | | Min FDE | | | Min ADE | | |
|---|---|---|---|---|---|---|---|---|---|---|---|---|
| | 40m | 80m | 120m | 40m | 80m | 120m | 40m | 80m | 120m | 40m | 80m | 120m |
| Continuous Diff. VLA | 0.72 | 1.53 | 2.71 | 0.35 | 0.74 | 1.00 | 0.35 | 0.76 | 1.44 | 0.21 | 0.44 | 0.61 |
| Discrete Diff. VLA | **0.67** | **1.37** | **2.19** | **0.34** | **0.69** | 1.02 | **0.29** | **0.68** | **1.06** | **0.17** | **0.39** | **0.59** |

# G  LIMITATIONS & FUTURE WORK

Here, we discuss our limitaitons and interesting future works.

- **Model Inputs.**  Our method relies on three-view images of the current frame as input. Nevertheless, single-frame images fail to capture velocity information, leaving the motion directions and speeds of surrounding vehicles unknown. Only by incorporating historical images and additional rich information as input can the model's interaction capabilities be fully utilized.

*Solution and future work:* We can incorporate historical images to enable the model to output not only planned trajectories but also the trajectories of key obstacles, providing a foundation for the reward model and subsequent trajectory game-theoretic interactions.

- **Reflection.**  First, Goal-Conditioned Generation should primarily focus on high-level objectives such as navigation compliance and traffic efficiency. In practical applications, scoring should prioritize these aspects. For rapid validation in this work, we directly adopted the PDM scorer without

task-specific adjustments. Second, in terms of Safety-Guided Regeneration, both the number of iterations and online inference attempts affect the final outcomes. While achieving better results requires sacrificing inference time, our experimental findings indicate that more inference opportunities do not necessarily yield better performance. Our analysis of failure cases reveals the following insights, as shown in Figure 6:

*1. Oscillation Between Boundaries:* The model tends to oscillate between boundary violations and collision avoidance in its final reasoning, particularly in scenarios with limited drivable space. This likely stems from increased difficulty caused by inherent errors in discrete trajectory representation. Future work could explore alternative methods to mitigate this issue.

*2. Navigation Correctness:* The reward function does not account for navigation correctness, leading to incorrect correction directions in certain scenarios. This can be addressed through iterative reward function refinement.

*3. Goal Point Selection:* Suboptimal goal point performance in specific scenarios limits correction capability when the search range is constrained. This could be improved by enhancing the base model through reinforcement learning or other advanced techniques.

*Solution and future work:* We can replace the rule-based reward with a model-based reward, and the search process can also be internalized within the model to some extent for reward-guided reflection, though this may introduce corner cases in certain scenarios.

• **Sample Efficiency.** Since the primary focus of this work is on method validation, we have not invested significant effort in algorithm optimization and acceleration, leaving substantial room for improvement.

*Solution and future work:* Since the output token count is relatively small, more inference iterations do not necessarily yield better results, and this could be reduced in future work. Additionally, engineering optimizations such as KV cache can be implemented to improve computational efficiency.

Overall, although some design choices may appear simple and certain limitations exist, we have thoroughly demonstrated the capabilities of ReflectDrive models for closed-loop planning in autonomous driving through extensive experiments. Moreover, we demonstrate the potential of ReflectDrive model to provide a safety driving behavior. It provides a high-performance, highly adaptable planner for autonomous driving systems.

## THE USE OF LARGE LANGUAGE MODELS (LLMS)

LLMs were used exclusively as writing assistance tools in preparing this manuscript. Specifically, we employed LLMs for grammar checking. All research ideation, experimental design, analysis, and scientific conclusions are entirely the work of the authors. The LLMs played no role in the conception of research questions, methodology development, or interpretation of results. Authors take full responsibility for all content in this paper, including any text refined with LLM assistance.

