# OpenReview forum: "Discrete Diffusion for Reflective Vision-Language-Action Models in Autonomous Driving"
_ICLR.cc/2026/Conference — ICLR 2026 Poster_

### Official Review · Reviewer_dnw7 · 2025-10-30

**Soundness:** 3
**Presentation:** 3
**Contribution:** 3
**Rating:** 6
**Confidence:** 5

**Summary:**

The paper proposes ReflectDrive, a VLA-based end-to-end planner that (1) discretizes 2D waypoints into a token codebook; (2) uses a discrete diffusion (masked-LM–style) model to inpaint trajectories; and (3) adds a reflection loop at inference: detect unsafe tokens via a safety scorer, do local discrete search to find a “safety anchor,” then re-inpaint around it—no gradients needed. Evaluated on NAVSIM with PDMS metrics (NC, DAC, TTC, Comfort, EP), the reflective loop substantially improves safety and progress, and with oracle agents it approaches human PDMS.

**Strengths:**

Pros:

1. **Clear, modular idea**: discrete tokenization → diffusion inpainting → gradient-free reflective editing. Nicely matches the structure of discrete diffusion (mask/remask/inpaint) and makes constraint injection tractable at inference.
2. **Practical safety loop**: the local Manhattan search over tokens + re-inpainting is simple and fast, with most cases finishing in 1–3 reflection iterations. With five diffusion steps, remask-low-confidence decoding, K=3 goals, NMS=0.9 m, max 10 refinement steps. Concrete enough to re-implement.
3. **Ablations and attribution**: Tables in the experiments section isolate contributions of goal-conditioning vs safety-guided regeneration;

**Weaknesses:**

Cons: (There are no obviously novel concerns, but some questions)
1. A core theme of the paper is safety and rules. From the ablation in Table 2, Goal-Conditioned Generation (GCG) and Safety-Guided Regeneration (SGR) have the larger impact. So I infer that the “safety” you highlight mainly comes from this reflective module. If so, on the one hand, where does Discrete Diffusion deliver unique value? On the other hand, did you try other trajectory parameterizations combined with goal and safety-anchor guidance?
2. There is a lack of description of the training data for multi-modal behavioral modes. I’m curious how the training data influences behavioral multi-modality: if the data is concentrated on a small set of behaviors (i.e., collected from only a few drivers’ styles), how do you train a top-k set of diverse behaviors? How do you avoid trajectory mode collapse?
3. There is a lack of description of the training data for multi-modal behavioral modes. I’m curious how the training data influences behavioral multi-modality: if the data is concentrated on a small set of behaviors (i.e., collected from only a few drivers’ styles), how do you train a top-k set of diverse behaviors? How do you avoid trajectory mode collapse?
4. In the Introduction, the paper emphasizes that guidance needs gradients, leading to slow sampling, hyper-parameter sensitivity, and potential instability. It repeatedly claims one benefit of discrete diffusion is eliminating gradient computation to reduce cost, and around line 60 states that other deployment schemes suffer slow sampling due to gradients. However, generation-based planning also depends on the number of sampling steps and the stochastic process; you use step-skipping with only 5 steps, yet the paper appears to lack experimental analysis on computational cost and speed, while stressing that “discretization enables efficient search.” The Limitations section explicitly notes that no algorithmic/engineering acceleration was applied and there is still room for efficiency improvement. Additionally, around line 203 the paper says accuracy is affected by discretization resolution; as resolution increases to meet accuracy/quality, inference speed will inevitably be impacted, and that speed is tightly coupled with the large model’s forward pass. I hope the authors can provide analysis and results validating the significance of discrete diffusion, the inference speed, and the effect of resolution.
5.  The paper calls NAVSIM a “real-world autonomous driving benchmark,” but NAVSIM is a data-driven simulator; that nuance matters for claims of real-world safety. Please clarify wording and discuss implications (non-reactive agents, sim-to-real).
 6. Some baselines use Camera+LiDAR (e.g., Transfuser, Hydra-MDP, DiffusionDrive/GoalFlow) whereas ReflectDrive is camera-only in the main line. Mixed-modality comparisons are informative but not apples-to-apples; add camera-only versions of diffusion/flow baselines or provide a C&L variant of ReflectDrive.
7. Safety oracle realism. The reflection loop’s default scorer assumes constant-velocity surrounding agents, which the authors themselves link to sub-par TTC/NC vs the oracle variant; this can bias safety judgments during reflection. Please add a learned (or stronger rule-based) predictor to close the realism gap.
8. Reward/metric shaping risks. The Global/Local scorers combine hard safety gates with weighted PDMS-like terms; optimizing a test-metric proxy at inference risks over-fitting to evaluator quirks (e.g., binary TTC thresholding). Provide robustness checks or alternative scorers.
9. Missing discretization specifics. The core idea hinges on codebook resolution (Δg), grid size |A|, and search radius δ. The paper states spatial range  [−100,100 ], but not Δg nor the token vocabulary size actually used at inference; please report these and study sensitivity.
10. Latency and systems profile.  The method is “designed for real-time,” yet no ms/frame or hardware stats are given (diffusion steps + reflection iterations + scorer costs). Please provide latency/throughput/memory on a target GPU/SoC and discuss worst-case bounds.

**Questions:**

1. Codebook & search: What Δg and codebook size |A| do you use? How is δ chosen (and its runtime impact)? Any adaptive δ strategy during reflection?
2. Noise schedule: You mention cosine-like masking, S=5 inference steps. What is the exact masking schedule during training/inference (token-wise remask policy, “remask low-confidence” threshold)?
3. Please formalize how “at-fault” collisions are computed and how TTC is thresholded; include ablations of scorer weights, W_ep, W_ttc, W_c.
4. Goal proposal: Why K=3 and d_NMS=0.9 m? Show sensitivity (K=1/3/5/10 already partly given—add per-metric trends and qualitative examples).
5. Variance & failures: In the oscillation failures (boundary vs collision), did larger δ or more iterations fix them, or is it a structural issue of discrete grids? Any hybrid continuous snap-back after the discrete anchor?

---

> ### Author Response · Authors · 2025-11-23
> **Response to Reviewer dnw7 [1 / 3]**
>
> We thank the reviewer for the positive comments. Regarding the concerns of the reviewer dnw7, we provide the following responses.
>
> > **W1. On the one hand, where does Discrete Diffusion deliver unique value? On the other hand, did you try other trajectory parameterizations combined with goal and safety-anchor guidance?**
>
> Our framework leverages discrete diffusion for effective trajectory search during reflection, as the discrete action space enables gradient-free and flexible optimization. Table 1 benchmarks our method against DiffusionDrive and Goalflow (methods that use goal and safety-anchor guidance), showing that our approach achieves better results.
>
>
>
> > **W2 & W3. There is a lack of description of the training data for multi-modal behavioral modes.?**
>
> We use the navtrain split from Navsim, which contains approximately 80k training samples. Indeed, training data significantly impacts multimodality. While training multimodal behavior models on small datasets is an interesting research question, companies in practice often possess extensive in-house data that significantly enhances generation quality. Furthermore, techniques like goal and anchor conditioning can be incorporated to avoid mode collapse.
>
> > **W4 & W10. Hope the authors can provide analysis and results validating the significance of discrete diffusion, the inference speed, and the effect of resolution.**
>
> We provide an ablation study on resolution in response to W9 & Q1. We evaluate the latency and memory profile of our method using a single NVIDIA H20. It is important to note that our current implementation is a research prototype based on a large VLM backbone (LLaDA-V) without engineering optimizations. The table below details the inference time breakdown. **The majority of time is consumed by the VLM backbone's forward pass (Trajectory Inpainting), while the scoring and search mechanisms are computationally negligible**.
>
> | Component | Time (s) | Note |
> | :--- | :--- | :--- |
> | **Stage 1: Goal-Conditioned Gen.** | **6.82 s** | **Total** |
> | - Goal Proposal | 0.62 s | Generation |
> | - Trajectory Inpainting | 6.06 s | 5 denoising steps |
> | - Scoring & Selection | 0.15 s | - |
> | **Stage 2: Safety-Guided Regen.** | **4.88 s** | **Per Iteration** |
> | - Safety Check | 0.13 s | - |
> | - Search | 0.84 s | - |
> | - Scoring | 1.68 s | - |
> | - Regeneration (Inpainting) | 2.14 s | 1 denoising step |
> | **Average Total Inference** | **~8.92 s** | - |
>
>
> - `Discretization enables efficient search.` The guidance mechanism requires gradient computation, which typically takes 1.5-2x longer than regeneration (as gradient calculation is significantly slower than the forward process). The computational overhead increases with model size. However, the search process requires only half the time of a forward pass. Moreover, by incorporating safety rules during the search, our method ensures trajectory safety—an advantage over guidance-based approaches, which may fail due to gradient errors or inappropriate guidance scales.
> - `Worst-Case Bounds` In the worst-case scenario (Safety-Guided Regeneration runs for the maximum 10 iterations), the total latency would reach approximately 55.66 s ($6.82 + 10 \times 4.88$). However, the average number of iterations required to resolve safety violations is only 0.43 (We calculated this average iterations on the NAVSIM test set), making the worst-case extremely rare.
> - `Memory Profile` The peak GPU memory usage is 57.1 GB. This high memory footprint is primarily due to the loading of the large VLM backbone and the lack of memory optimization techniques in this prototype phase.
>
> > **W5. Please clarify wording and discuss implications (non-reactive agents, sim-to-real) of Navsim**
>
> Navsim is considered a real-world autonomous driving benchmark because it utilizes real-world driving data, and this claim is widely recognized in existing literature [1,2]. While non-reactive agents and the sim-to-real gap may affect evaluation accuracy, Navsim remains the most suitable benchmark for assessing end-to-end methods. This is because world models struggle to provide reliable evaluation, and simulation platforms like CARLA face more significant sim-to-real gaps.
>
> [1] Dauner, Daniel, et al. "Navsim: Data-driven non-reactive autonomous vehicle simulation and benchmarking." Advances in Neural Information Processing Systems 37 (2024): 28706-28719.
>
> [2] Liao, Bencheng, et al. "Diffusiondrive: Truncated diffusion model for end-to-end autonomous driving." Proceedings of the Computer Vision and Pattern Recognition Conference. 2025.

---

> ### Author Response · Authors · 2025-11-23
> **Response to Reviewer dnw7 [2 / 3]**
>
> > **W6. Mixed-modality comparisons are informative but not apples-to-apples; add camera-only versions of diffusion/flow baselines or provide a C&L variant of ReflectDrive.**
>
> Multimodal input can indeed enhance performance, as LiDAR provides more precise 3D information. However, we did not use LiDAR in this work to maintain compatibility with our VLM pretraining. While incorporating LiDAR into VLMs represents an interesting research direction, it falls outside the scope of our current study. By relying solely on camera inputs, our method still surpasses existing baselines, demonstrating the effectiveness of our approach.
>
>
> > **W7. Please add a learned (or stronger rule-based) predictor to close the realism gap.**
>
> In Table 1, we present the results of using a privileged ground-truth oracle for reflection. This approach achieves better performance than using an unreliable reward. In practice, more sophisticated learned or rule-based predictors with additional engineering designs could be employed to further improve performance. However, this paper primarily focuses on demonstrating the effectiveness of our proposed framework. We thank the reviewer for pointing this out. We leave the exploration of this aspect for future work due to time constraints during the rebuttal period.
>
> > **W8 & Q3. Provide robustness checks or alternative scorers.**
>
> - We follow the NAVSIM benchmark to compute the at-fault collision and TTC metrics. The at-fault collision is defined as an event where the ego vehicle's bounding box intersects with another agent's bounding box. TTC is calculated as the time required for the ego vehicle and another agent to collide if both maintain their current speeds and headings.
> - We conducted a sensitivity analysis on the scorer weights ($W_{ep}, W_{ttc}, W_c$) and the binary TTC threshold. The evaluation metrics were kept unchanged throughout this analysis. As shown in the table below, our method demonstrates strong robustness to hyperparameter variations:
>
> | W_ep        | W_ttc | W_c | TTC Thresh (s) | PDM Score |
> |-------------|-------|------|----------------|-----------|
> | 5  | 5     | 2    | 1.0          | 91.1    |
> | 5           | 5     | 0    | 1.0            | 91.1      |
> | 2           | 5     | 5    | 1.0            | 91.1      |
> | 0           | 5     | 5    | 1.0            | 91.2  |
> | 5           | 2     | 5    | 1.0            | 91.1      |
> | 5           | 0     | 5    | 1.0            | 90.9      |
> | 5           | 5     | 2    | 0.5            | 90.0      |
> | 5           | 5     | 2    | 1.5            | 91.0      |
>
> > **W9 & Q1. Missing discretization specifics.**
>
> - `Grid Resolution & Codebook` We utilize a grid resolution of $\Delta g = 0.3$ meters. Given the spatial range of $[-100m, 100m]$ defined in our training configuration (Table 4), this results in a codebook size of $|A| \approx 667$ tokens per dimension, these all training configurations.
> - `Search Radius ($\delta$)` During the Search, we use a fixed Manhattan distance of $\delta \le 10$ grids (approx. 3 meters), which is a fixed value. The search radius of 10 is sufficient to nudge the vehicle out of collision zones without deviating too far from the original intent. We have already provide ablation results in Figure4 \(c\).
> - `Ablation on resolution` We investigate the impact of grid resolution $\Delta g$ on performance. The model exhibits robust high performance ($>90.7$) across the range of 0.2m to 0.5m. Using an excessively fine granularity ($\Delta g = 0.1$m) leads to a significant performance drop (88.2). This is likely due to the drastically increased codebook size ($|A|=2000$), which complicates the classification task for the diffusion model. It's interesting to find that high resolution doesn't result in performance degradation due to discretization. To further verify this phenomenon, **scaling up with in-house data, we found that discrete diffusion surpassed continuous diffusion, yielding better performance (see General Response)**.
>
> | Resolution ($\Delta g$) | 0.1 m | 0.2 m | 0.3 m | 0.4 m | 0.5 m |
> | :--- | :--- | :--- | :--- | :--- | :--- |
> | **PDMS Score** | 88.2 | 91.3 | 91.1 | 90.7 | 91.2 |
>
> > **Q2. Noise schedule details.**
>
> We utilize a linear noise schedule. During training, we sample mask ratios uniformly. During inference, we accelerate the process to $S=5$ steps. At each step, we perform parallel decoding: the model predicts the probabilities for all masked tokens, and we permanently unmask (fix) the top portion of tokens with the highest confidence scores, such that the number of masked tokens decreases linearly to zero over the 5 steps.

---

> > ### Comment · Reviewer_dnw7 · 2025-11-25
> >
> > Thank you for the explanation. I will retain my original score and keep high confidence. Best of luck.

---

> > > ### Author Response · Authors · 2025-11-25
> > >
> > > Dear Reviewer dnw7,
> > >
> > > We sincerely thank you for your insightful feedback and support. Your constructive comments have been instrumental in refining our work!
> > >
> > > Best regards,
> > >
> > > The Authors

---

> ### Author Response · Authors · 2025-11-23
> **Response to Reviewer dnw7 [3 / 3]**
>
> > **Q4. Goal proposal sensitivity (K=3, d_NMS=0.9).**
>
> During rebuttal, we supplemented the corresponding ablation and visualization results (Fig. 7&8). While this increased diversity improves the Goal-Conditioned Generation score, it also introduces higher variance that can challenge the subsequent refinement stage. In practice, we selected a smaller number of goals ($K=3$) from an efficiency perspective, requiring them to maintain a certain distance, and chose $d_{\text{NMS}}=0.9$ was determined through visualization rather than quantitative scores, as this value adequately captured multi-modal behavior without requiring excessive trajectory separation.
>
> | Goal Points ($K$) | 1 | 3  | 5 |  10 |
> | :--- | :--- | :--- | :--- | :--- |
> | **PDMS** | 84.1 | 87.4  | 88.7 |  87.6 |
>
> | NMS Threshold (m) | 0.0 | 0.3 | 0.9 | 1.2 |
> | :--- | :--- | :--- | :--- | :--- |
> | **PDMS** | 85.8 | 86.5 | 87.4 | 88.0 |
>
>
> > **Q5. Variance & failures (Oscillation).**
>
> As discussed in the Limitations section (Appendix G), we have observed cases of "Oscillation Between Boundaries," particularly in narrow drivable areas. This occurs when the discrete grid points around a boundary all carry penalty scores. Increasing the search radius $\delta$ helps in some cases, but we believe a solution involves refining the reward function to include a continuous "distance-to-boundary" gradient rather than a binary threshold.

---

### Official Review · Reviewer_pLbU · 2025-10-31

**Soundness:** 2
**Presentation:** 3
**Contribution:** 2
**Rating:** 4
**Confidence:** 5

**Summary:**

The paper presents ReflectDrive, a novel framework that integrates a reflection mechanism for safe trajectory generation in autonomous driving using discrete diffusion models. The authors aim to enhance the safety and reliability of trajectory planning by combining discrete diffusion with goal-based trajectory planning. The framework introduces three scoring functions—Global Scorer, Safety Scorer, and Local Scorer—to evaluate the generated trajectories and ensure adherence to safety constraints.

**Strengths:**

1. **Safety Focus**: The emphasis on safety through the reflection mechanism and scoring functions demonstrates a commitment to developing reliable autonomous systems.
2. **Evaluation on Closed-loop Benchmarks**: The use of NAVISM benchmark for evaluation adds credibility to the findings and demonstrates practical applicability.
3. **Easy to follow**: The key idea is clear and the paper is easy to understand.

**Weaknesses:**

1. **Limited Novelty**: The primary contribution of the paper appears to be the combination of discrete diffusion models with goal-based trajectory planning. This concept closely resembles the existing GoalFlow work, where flow matching is replaced by discrete diffusion, and more scorers are used. The incremental nature of this contribution raises questions about its originality.

2. **Insufficient Ablation Studies**: While the paper proposes three distinct scoring functions, comprehensive ablation studies are lacking. Only two of the scorers are evaluated, which does not provide a complete understanding of the impact of each component on the overall performance. A thorough ablation analysis is necessary to validate the effectiveness of all proposed components.

3. **Lack of Detail in Safety-Guided Regeneration**: The section on Safety-Guided Regeneration is overly simplistic. It does not adequately address the success rate of the local search strategy or provide contingency plans for search failures. Additionally, the generalizability of this search method in extreme or unusual scenarios is not explored, which is critical for assessing its robustness.

**Questions:**

which scorer play the most important role? Is it possible to use single scorer?

---

> ### Author Response · Authors · 2025-11-23
> **Response to Reviewer pLbU**
>
> We thank the reviewer for the constructive comments. Regarding the concerns of the reviewer pLbU, we provide the following responses.
>
> > **W1. Limited Novelty. The primary contribution of the paper appears to be the combination of discrete diffusion models with goal-based trajectory planning.**
>
> - The reviewer seems to have overlooked our key contribution: **the integration of safety rules into trajectory generation**. This approach directly faces the critical challenge of enforcing hard safety constraints in imitation learning—**a novelty also recognized by Reviewers qxE7 and dnw7**.
> - Furthermore, our framework leverages discrete diffusion for effective trajectory search during reflection, while goal-conditioned generation supplies multimodal initial trajectories. **These coordinated components establish a solid foundation for the reflection mechanism**.
> - It is also noteworthy that when scaling up with in-house data, discrete diffusion outperformed continuous diffusion models, a finding we consider valuable to the research community (see General Response).
>
>
>
> > **W2. Insufficient Ablation Studies. Only two of the scorers are evaluated, which does not provide a complete understanding of the impact of each component on the overall performance.**
>
>
> In Table 2, the "Safety-Guided" method employs two score functions that must be used together during the search process, while "Goal-Cond" utilizes only one score function, as described in Section 4.2. We have therefore already conducted an ablation study for each component of our method. If reviewer is interested, we provide more ablations about parameter choice in our methods in Appendix D.
>
> > **W3. Lack of Detail in Safety-Guided Regeneration.**
>
>
> - `Largely replaces the rule-based refinement` By employing a local search and leveraging the model's inpainting capabilities, the final trajectory is still generated by the model and benefits from the scalability of learning-based methods. **In fact, we can significantly simplified the complex rule-based refinement process**. The results in Tables 1 and 2 alos demonstrate the effectiveness of this approach. Thus, describing our method as overly simplistic is not accurate.
> - `Contingency Plans` We set a maximum budget of 10 iterations to ensure real-time feasibility. In the rare event that the algorithm reaches this limit without finding a fully safe trajectory, the system outputs the candidate with the highest safety score found during the search. **We provide only a reflection framework; in real-world deployment, more engineering designs can be incorporated to establish a fallback system**.
> - `Generalizability` The mechanism's robustness has been validated in challenging scenarios involving large-angle turns and dense dynamic interactions, as visualized in Figure 3. While we identified some "oscillation" cases in constrained spaces (Appendix G), these represent edge cases that we plan to address with continuous reward shaping in future work. We would be happy to test more extreme or unusual scenarios in NAVSIM if the reviewer could provide further details about the specific scenarios they would like to see evaluated.
>
> > **Q1. Which scorer play the most important role? Is it possible to use single scorer?**
>
>
> "Safety Scorer" is the most important component, serving as the foundation for local search. It is possible to use this single scorer if the model-generated trajectory is of sufficiently high quality, allowing us to skip goal-conditioned generation and perform only a single-step search. High-quality trajectory generation has been demonstrated to be feasible through the use of large-scale in-house data, enabling the production of reliable and effective results (see General Response).

---

### Official Review · Reviewer_qxE7 · 2025-11-01

**Soundness:** 2
**Presentation:** 4
**Contribution:** 3
**Rating:** 4
**Confidence:** 5

**Summary:**

This paper proposes ReflectDrive, a framework for end-to-end autonomous driving that leverages a Vision-Language-Action (VLA) model with a discrete diffusion planner. The core contributions are: 1) The discretization of the continuous driving space into an action codebook, which allows for the use of a discrete diffusion model for trajectory generation. 2) A reflection mechanism with a gradient-free, iterative inference-time process. This mechanism first generates multi-modal trajectories based on sampled goals and then uses a safety-guided regeneration step to correct unsafe waypoints identified by a safety scorer.

**Strengths:**

1. The application of discrete diffusion and an inpainting-based reflection mechanism to a VLA model for driving is a novel approach to integrating safety constraints.
2. The work directly tackles the critical and unsolved problem of enforcing hard safety constraints (like collision avoidance and drivable area compliance) within imitation learning-based frameworks.

**Weaknesses:**

1. The safety score is a very dedicated design that still falls into the rule-based refinement. More critically, obtaining these scores would inevitably employ some privileged information. For example, the DAC score will require the knowledge of whether the car will fall out of the drivable region, which is impossible to reliably obtain purely from sensor data in the real world. This reliance on an oracle-like safety scorer questions the practical applicability of the reflection mechanism. It will be nice to see ablation studies where some less reliable, learning-based rewards are involved.
2. To justify the effectiveness of the discrete action space, it would be beneficial to see an ablation study on different granularities of the discretization.

**Questions:**

1. What do the variables c and s represent in Equation (2)?
2. The safety scoring mechanism described in Appendix C (e.g., $m_{DAC}$, $m_{NC}$) seems to require ground-truth or privileged simulation information. How do you envision this safety-guided regeneration being applied in a real-world vehicle that relies only on on-board sensor data?

---

> ### Author Response · Authors · 2025-11-23
> **Response to Reviewer qxE7**
>
> We thank the reviewer for the constructive comments. Regarding the concerns of the reviewer qxE7, we provide the following responses.
>
> > **W1 & Q2. The safety score still falls into the rule-based refinement. Obtaining these scores would employ privileged information.**
>
> - We used privileged information **only to validate our reflection mechanism**, rather than to train a model for prediction or detection, as the latter is beyond the scope of this paper. Our key contribution is the integration of safety rules into trajectory generation—a task that has been challenging for prior learning-based methods. **Even with less reliable constant-velocity predictions (see Table 1), our model shows significant improvement, highlighting the value of the reflection mechanism**.
> - In real-world applications, we can leverage well-developed prediction and detection modules. These commercially proven modules can be reused with no additional development. **This makes the reflection mechanism applicable in practice**. Furthermore, engineering optimizations—such as focusing solely on near-field collisions and boundary violations—can further enhance the accuracy of lane detection and object detection in this context.
> - Notably, the trajectory is still generated by the diffusion model, leveraging its inpainting capability rather than rule-based refinement. We thank the reviewer for the important suggestion on learning-based rewards, which we agree is a crucial direction for future work as we build toward a fully learning-based system.
>
>
> > **W2. It would be beneficial to see an ablation study on different granularities of the discretization .**
>
> - We investigate the impact of grid resolution $\Delta g$ on performance. The model exhibits robust high performance ($>90.7$) across the range of 0.2m to 0.5m. Using an excessively fine granularity ($\Delta g = 0.1$m) leads to a significant performance drop (88.2). This is likely due to the drastically increased codebook size ($|A|=2000$), which complicates the classification task for the diffusion model.
>
> | Resolution ($\Delta g$) | 0.1 m | 0.2 m | 0.3 m | 0.4 m | 0.5 m |
> | :--- | :--- | :--- | :--- | :--- | :--- |
> | **PDMS Score** | 88.2 | 91.3 | 91.1 | 90.7 | 91.2 |
>
> - It's interesting to find that high resolution doesn't result in performance degradation due to discretization. To further verify this phenomenon, **scaling up with in-house data, we found that discrete diffusion surpassed continuous diffusion, yielding better performance (see General Response)**.
>
>
>
> > **Q1.	What do the variables c and s represent in Equation (2)?**
>
> Here, $s$ is the mask ratio, and $c$ represents the scene context (e.g., multi-view images, language instruction, ego state), as mentioned in Section 3.2. We have made further revisions to the main text to avoid misunderstandings.

---

### Author Response · Authors · 2025-11-23
**General Response**

We thank all the reviewers for the effort engaged in the review phase and the constructive comments.

We have revised our paper (highlighted in blue text color). The modifications are summarized as follows.

1. **(For Reviewer qxE7, dnw7, pLbU)**. We have added a scaled-up in-house data experiment in Appendix F to demonstrate the effectiveness of discrete diffusion.
2.  **(For Reviewer qxE7, dnw7)**. We have added a detailed ablation study on discretization granularity in Appendix D.1.
3. **(For Reviewer qxE7)**. We have added the definitions of variables $c$ and $s$ in Section 3.2.
4.  **(For Reviewer dnw7)**. We have included a  sensitivity analysis of Goal Proposal Parameters ($K$ and $d_{\text{NMS}}$) in Appendix D.3,  along with an inference latency and system profile analysis in Appendix C. Additionally, inference details have been added to Section 3.2, while the grid resolution and codebook size are now specified in Section 4.1.
7.  **(For Reviewer dnw7, pLbU)**. We have added a robustness check for Scoring Functions in Appendix D.2.
9.  **(For Reviewer pLbU, dnw7)**. We have clarified the fallback strategy for regeneration in Section 4.2.


> **Effectiveness of Discrete Diffusion**

To validate the effectiveness of our framework, we use in-house  dataset about 1 billion samples which is more comprehensive and owns a larger scale and richer scenario diversity. We compared the open-loop performance metrics between our discrete diffusion VLA and continuous diffusion VLA. All models generated 8 trajectories, where "Top1" refers to the highest-confidence trajectory, and "Min" indicates the trajectory with the minimum error. The specific values are shown below, demonstrating the effectiveness of using discrete representations in trajectory generation.

| Methods | top1 40m fde | top1 80m fde | top1 120m fde | top1 40m ade | top1 80m ade | top1 120m ade | min 40m fde | min 80m fde | min 120m fde | min 40m ade | min 80m ade | min 120m ade |
|---|---|---|---|---|---|---|---|---|---|---|---|---|
| Continue Diffusion VLA| 0.72 | 1.53 | 2.71 | 0.35 | 0.74 | 1.0 | 0.35 | 0.76 | 1.44 | 0.21 | 0.44 | 0.61 |
| Discrete Diffusion VLA | 0.67 | 1.37 | 2.19 | 0.34 | 0.69 | 1.02 | 0.29 | 0.68 | 1.06 | 0.17 | 0.39 | 0.59 |

---

### Author Response · Authors · 2025-12-03
**Rebuttal Summary**

Dear Area Chair,

To facilitate your assessment, we summarize the current status of our submission below.

We thank all reviewers for their constructive feedback. We are encouraged that **Reviewer dnw7 (Score: 6)** maintained their positive rating, explicitly praising our "clear, modular idea" and "practical safety loop." Crucially, multiple reviewers (qxE7, pLbU) acknowledged that our reflection mechanism addresses a fundamental bottleneck in autonomous driving: the difficulty of enforcing hard safety constraints within end-to-end Imitation Learning. We believe we have fully addressed the remaining concerns regarding validation and novelty through significant new experiments.

`1. Superiority of Discrete Diffusion on Billion-Scale Data`
To conclusively validate our design, we conducted a large-scale experiment using an in-house dataset of approximately **1 billion samples**. The results show that our Discrete Diffusion VLA consistently outperforms the Continuous Diffusion VLA baseline across all metrics (e.g., reducing Top-1 120m FDE from 2.71 to 2.19). **This empirical evidence confirms that discretizing the action space is not merely a constraint, but a superior design choice for large-scale trajectory generation.**

`2. Significance of Reflection in Imitation Learning`
Reviewers identified that our core contribution lies in integrating safety rules into generation—a task previously challenging for learning-based methods. By leveraging the masking capability of discrete diffusion, we enable the model to "reflect" and self-correct based on safety scores without expensive gradient-based guidance. **This confirms our framework successfully bridges the gap between data-driven imitation and rule-based safety requirements.**

`3. Robustness and Real-World Feasibility`
Addressing concerns about sensitivity and latency, our new ablation studies demonstrate that the model maintains high performance across a wide range of grid resolutions (0.2m–0.5m) and scorer weights. We also clarified that the inference latency is primarily dominated by the VLM backbone, while our discrete search adds negligible overhead. **These results verify that our method is robust to hyperparameter changes and efficient enough for practical deployment.**

Based on these substantial additions—especially the validation on billion-scale data—we believe the paper is strengthened.


Best regards,

The Authors

---

### Meta-Review · Area_Chair_B5mN · 2025-12-30

**Summary:**

This paper proposes a new method for end-to-end autonomous driving using Vision-Language-Action (VLA) models. Its technical core contributions evolve a scheme for discretizing the continuous driving space into an action codebook that enables applying discrete diffusion models and an inference-time reasoning mechanism that first generates trajectories and subsequently refines them using a safety-aware reasoning mechanism.  The method is evaluated on the NAVSIM benchmark, demonstrating advantages in safety-critical trajectory generation.

**Reviewer Concerns:**

The reviewers had three main types of concerns. First and probably the largest concern is the potential lack of novelty of the work (pLBU). Second, it does not sufficiently ablate some of the design choices (plbU, qxE7). Third, there were several concerns about lack of details or imprecise formulations (pLbU, dnw7).

**Reviewer Scores:**

The authors addressed most of the minor points (raised by dnw7) satisfactorily. They also pointed out that some of the requested ablations already exist and provided additional ablation on the granularity of the discretization. It is probably hard to judge to which extent reviewer pLBU would find the response on the novelty satisfactory (e.g., there are other, albeit older, works integrating safety rules in trajectory generation, which the authors mention as a contribution in their response). On the other hand, the integration in the present setting may be of interest anyway.

---

### Decision · Program_Chairs · 2026-01-26

Accept (Poster)